# Telencephalic outputs from the medial entorhinal cortex are copied directly to the hippocampus

**Sau Yee Tsoi**[1†]**, Merve Öncül**[1†]**, Ella Svahn**[1‡]**, Mark Robertson**[1§]**, Zuzanna Bogdanowicz**[1]**, Christina McClure**[1#]**, Gülşen Sürmeli**[1,2*]

[1]University of Edinburgh, Centre for Discovery Brain Sciences, Edinburgh, United Kingdom; [2]Simons Initiative for the Developing Brain, University of Edinburgh, Edinburgh, United Kingdom

**\*For correspondence:**
gsurmeli@ed.ac.uk

[†]These authors contributed equally to this work

**Present address:** [‡]University College London, London, United Kingdom; [§]NHS Greater Glasgow and Clyde, Glasgow, United Kingdom; [#]VectorBuilder Inc, Chicago, United States

**Abstract** Complementary actions of the neocortex and the hippocampus enable encoding and long-term storage of experience dependent memories. Standard models for memory storage assume that sensory signals reach the hippocampus from superficial layers of the entorhinal cortex (EC). Deep layers of the EC on the other hand relay hippocampal outputs to the telencephalic structures including many parts of the neocortex. Here, we show that cells in layer 5a of the medial EC send a copy of their telencephalic outputs back to the CA1 region of the hippocampus. Combining cell-type-specific anatomical tracing with high-throughput RNA-sequencing based projection mapping and optogenetics aided circuit mapping, we show that in the mouse brain these projections have a unique topography and target hippocampal pyramidal cells and interneurons. Our results suggest that projections of deep medial EC neurons are anatomically configured to influence the hippocampus and neocortex simultaneously and therefore lead to novel hypotheses on the functional role of the deep EC.

## Editor's evaluation

In this work, the authors combine a variety of novel circuit mapping techniques to characterize a novel projection pathway from layer 5a neurons in the medial entorhinal cortex to region CA1 of the hippocampus. By utilizing cell specific viral labelling techniques, RNA sequencing based projection mapping, and optogenetic-aided in vitro physiology, the authors show evidence that the same neurons in layer 5a of medial entorhinal cortex project to both cortical areas of the telencephalon and the hippocampus. This work raises the possibility that deep layers of the entorhinal cortex coordinate hippocampal-cortical interactions. The manuscript will be of interest to readers in the field of anatomy and hippocampal physiology.

## Introduction

Interplay between the hippocampus and the neocortex is a central tenet of theories for systems memory consolidation (**Marr, 1971**; **McClelland et al., 1995**). The entorhinal cortex (EC) mediates the two-way communication between the neocortex and the hippocampus. Its superficial and deep layers, respectively, channel inputs to and outputs from the hippocampus. During a new experience the superficial layers relay convergent inputs from the cortex to the hippocampus where initial conjunctive representations of everyday experiences are generated. Subsequently during rest or sleep, hippocampal output is redistributed to the brain (**Squire et al., 2015**; **Frankland and Bontempi, 2005**). Leading theoretical models consider the deep layers of EC as a relay of hippocampal output that

reinstates neocortical activation patterns representing memories, enabling neocortex-wide synaptic changes required for long-term storage (*Koster et al., 2018*; *Kumaran et al., 2016*; *McClelland et al., 1995*; *Schapiro et al., 2017*).

Recent discoveries revealed complex and distinct input–output interactions of molecularly defined subtypes of neurons in the deep layers of medial EC (MEC), suggesting their functions extend beyond passing on hippocampal outputs. Layers 5a (L5a) and 5b (L5b) of the EC are genetically distinct with differential input–output organizations. L5a but not L5b is the sole origin of the long-range telencephalic outputs of the EC (*Sürmeli et al., 2015*), whereas L5b, but not L5a, is directly influenced by superficial layers of EC (*Beed et al., 2020*; *Sürmeli et al., 2015*). Inputs from the dorsal hippocampus preferentially target L5b over L5a (*Rozov et al., 2020*; *Sürmeli et al., 2015*; *Wozny et al., 2018*) whereas ventral hippocampus preferentially targets L5a (*Rozov et al., 2020*). Moreover, the capacity of deep layers to integrate neocortical inputs with hippocampal output (*Czajkowski et al., 2013*; *Beed et al., 2020*; *Sürmeli et al., 2015*), suggest more complex roles.

An intriguing possibility, noted in experiments with classic retrograde tracers, is that deep layers of MEC might also project to the hippocampal CA1 region, which provides the deep EC with hippocampal output (*Köhler, 1985*; *Witter and Amaral, 1991*). Such recurrent connectivity between CA1 and the deep MEC could be important in coordinating hippocampal–entorhinal activity during memory consolidation (*Ólafsdóttir et al., 2016*), provide a shortcut for hippocampal outputs to re-enter the hippocampus (*Kumaran and McClelland, 2012*) or have a more global coordination function encompassing the hippocampus and the neocortex (*Khodagholy et al., 2017*). Despite its potential functional significance, the key properties of this projection pathway including whether it shares the same source with the telencephalon projections and which neurons it targets in the hippocampus remain unknown.

Here we reveal that, neurons in L5a, but not L5b, of the MEC project to hippocampal CA1. We establish that the telencephalon-projecting neurons copy their outputs directly to pyramidal and subclasses of interneuron populations in CA1. Our data define a novel anatomical framework for layer 5a of the MEC to coordinate neocortical and hippocampal networks.

## Results

To find out whether neurons in either L5a or L5b project to the hippocampus we injected retrograde adeno-associated viral vectors (AAVs) expressing green fluorescent protein (GFP) or mCherry into intermediate hippocampus (*Figure 1A, B*; *Figure 1—figure supplement 1A,B*). As well as finding retrogradely labelled cell bodies in layer 3 of MEC, which contains neurons that are well established as a critical input to stratum lacunosum-moleculare of CA1 (*Naber et al., 2001*; *Kitamura et al., 2014*), we also found labelled cell bodies in L5a, which we identified by immunostaining against the ETS variant transcription factor-1 (Etv1) protein. Retrogradely labelled neurons were not found in L5b.

To further investigate this projection, we identified and validated a transgenic mouse line that enables specific genetic access to L5a. In this mouse line, Cre expression is driven by the promoter of the retinol binding protein four gene (*Rbp4-Cre*; *Gerfen et al., 2013*). Following injection of a Cre-dependent AAV expressing GFP or mCherry into the deep MEC of *Rbp4-Cre* mice, fluorescent signal was detected mainly in L5a (*Figure 1C, D* and *Figure 1—figure supplement 1C*). At the centre of the injection site, where labelling was densest, the majority of neurons in L5a (71.3% ± 2.2 %, *n* = 3 mice, 180 neurons) were found to be fluorescently labelled (*Figure 1D*). Labelled neurons were non-GABAergic (99.5% ± 0.3%, *n* = 3 mice, 1137 cells; *Figure 1—figure supplement 1D*), most of which expressed Etv1 (88.6% ± 3.1 %, *n* = 3 mice, 1866 cells; *Figure 1D*). This is similar to the proportion of Etv1 expression in all L5a neurons (*Sürmeli et al., 2015*) and also in retrogradely labelled subpopulations of L5a neurons from the hippocampus and from cortical and subcortical target areas (*Figure 1—figure supplement 1E–H*). Moreover, axons of L5a neurons labelled in the *Rbp4:Cre* line were found in the telencephalon (*Figure 1—figure supplement 2*). Thus, we established that the *Rbp4-Cre* line provides unbiased genetic access to excitatory projection neurons in L5a.

To what extent does the projection from L5a to the hippocampus differ from previously identified projections arising from superficial layers of MEC? In contrast to projections from superficial layers 2 and 3, which, respectively, target stratum lacunosum (SL) and stratum moleculare (SM) of CA1 (*Kitamura et al., 2014*), axons from L5a neurons were primarily found in deep stratum pyramidale (SP) and SL, but were sparse in SM (*Figure 2A–D* and *Figure 2—figure supplement 1A*). Axons from L5a also

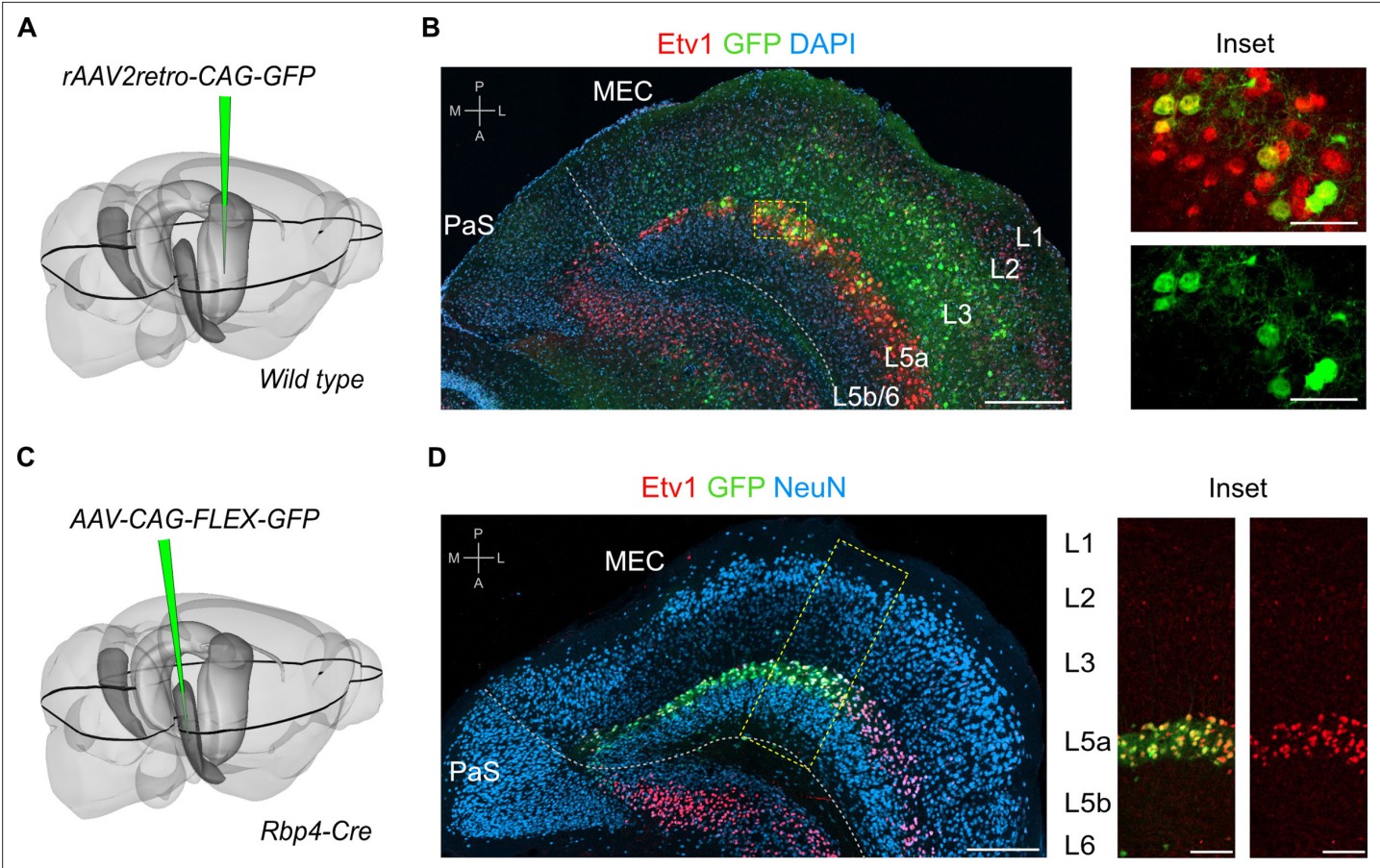

**Figure 1.** L5a but not layer 5b of the medial entorhinal cortex (MEC) projects to the hippocampus. (**A**) Schematic showing the injection site of a retrograde adeno-associated viral vector (AAV) in the intermediate hippocampus (light grey) of wild-type mice. The black outline in the horizontal plane roughly corresponds to the dorso-ventral position of the horizontal section shown in B. Corresponding injection site image is shown in *Figure 1— figure supplement 1A*. (**B**) Immunostained brain section in the horizontal plane showing retrogradely labelled neurons in L3 and L5a of the MEC. ETS variant transcription factor-1 (Etv1) expression marks L5a (scale bar: 250 μm). Inset: high-magnification image of boxed region in left panel showing co-expression of green fluorescent protein (GFP) and Etv1 (scale bar: 50 μm). PaS, parasubiculum; P, posterior; A, anterior; M, medial; L, lateral. (**C**) Schematic showing the injection site of an AAV in the deep MEC (dark grey) of *Rbp4-Cre* mice for Cre-dependent expression of GFP. The black outline in the horizontal plane roughly corresponds to the dorso-ventral position of the slice shown in D. (**D**) Entorhinal area on an immunostained brain section in the horizontal plane showing the overlapping expression of GFP with Etv1 (scale bar: 250 μm). Inset: high magnification of boxed region in middle panel showing GFP and Etv1 expression confined mainly to L5a in MEC (scale bar: 100 μm).

The online version of this article includes the following figure supplement(s) for figure 1:

**Figure supplement 1.** Injection site images, quantification of co-localization between retrograde markers and ETS variant transcription factor-1 (Etv1).

**Figure supplement 2.** L5a neurons of the medial entorhinal cortex (MEC) innervate neocortical and subcortical areas.

differed in their proximo-distal organization within CA1. Whereas projections from superficial layers of MEC favour proximal CA1 (*Tamamaki and Nojyo, 1995*), axons from L5a had highest density in distal CA1 (paired one-tailed Wilcoxon signed rank test, p = 0.0312, *n* = 5 mice; *Figure 2E–H* and *Figure 2—figure supplement 1B–F*) and the subiculum (*Figure 2I* and *Figure 2—figure supplement 1G*). We observed the same topography when axons and axon terminals of L5a neurons were labelled with membrane-bound GFP and fluorescently tagged presynaptic proteins (synaptophysin), confirming that the topography is not a result of labelling of passing axons (*Figure 2J–O*). This distinct topography suggests a unique functional role for the projections from L5a of MEC to CA1.

Our findings suggest two intriguing possibilities for the organization of output neurons in L5a of the MEC. The hippocampus-projecting neurons in L5a may be distinct from the telencephalon-projecting neurons, implying separate processing within the deep MEC layers of signals to the hippocampus and telencephalon. Alternatively, the same neurons in L5a may project to both the telencephalon and the

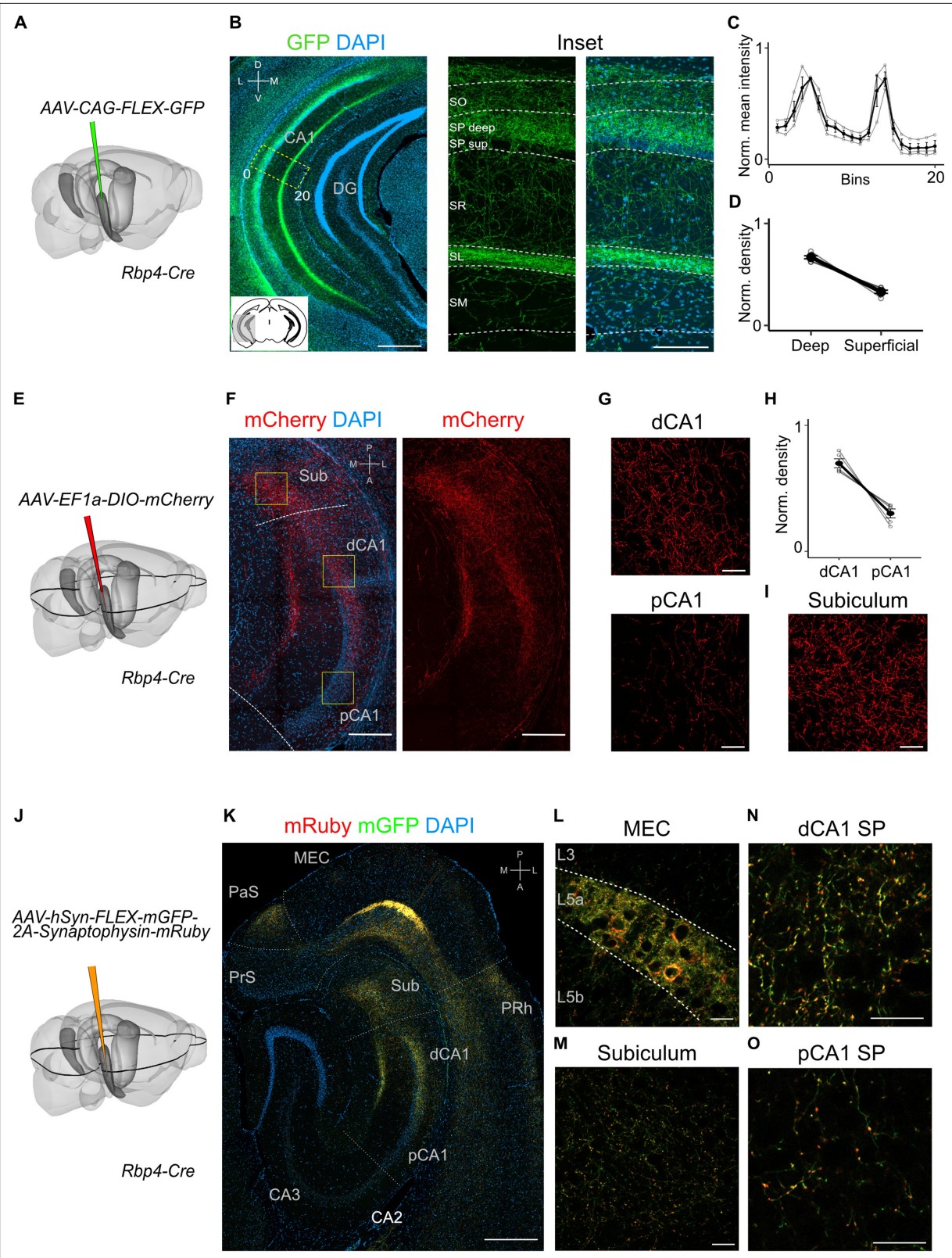

**Figure 2.** Projections to CA1 of L5a neurons of the medial entorhinal cortex (MEC) have distinct topography. (**A**) Schematic showing the injection strategy of a Cre-dependent virus in the deep MEC of *Rbp4-Cre* mice. (**B**) Immunostained coronal brain section showing the distribution of axons of MEC L5a cells in CA1 (scale bar: 500 μm). Inset: high-magnification image of boxed region in left panel (scale bar: 100 μm). Bottom-left schematic indicates the antero-posterior position of the slice. Corresponding injection site image is shown in ***Figure 2—figure supplement 1A***. SO, stratum

*Figure 2 continued on next page*

*Figure 2 continued*

oriens; SP, stratum pyramidale; SR, stratum radiatum; SL, stratum lacunosum; SM, stratum moleculare; D, dorsal; V, ventral; M, medial; L, lateral. (**C**) Quantification of fluorescence intensity of green fluorescent protein (GFP) across the layers of CA1. The area between 0 and 20 in (**B**) was divided into 20 equally sized bins, and the mean fluorescence intensity was calculated for each bin. Values shown are normalized mean fluorescence intensity values (*n* = 6 mice, 2 sections per brain). (**D**) Normalized mean fluorescence density values in deep versus superficial SP (*n* = 6 mice). Within SP, axon density in deep SP was significantly higher than density in superficial SP (paired one-tailed Wilcoxon signed rank test, p = 0.0156, *n* = 6). (**E**) Schematic showing the injection strategy of a Cre-dependent virus in the deep MEC of *Rbp4-Cre* mice. The black outline in the horizontal plane roughly corresponds to the dorso-ventral position of the horizontal slice shown in (**F**).( **F**) Immunostaining of a horizontal brain section showing the distribution of axons of MEC L5a cells in the subiculum (Sub), distal CA1 (dCA1), and proximal CA1 (pCA1; scale bar: 200 μm). Corresponding injection site image is shown in *Figure 2—figure supplement 1B*. (**G**) High-magnification images of the pyramidal cell layer in distal (top) and proximal (bottom) CA1 showing axonal labelling (scale bar: 20 μm). (**H**) Normalized mean fluorescence density values in distal versus proximal CA1 (*n* = 5 mice). (**I**) High-magnification image of the subiculum showing axonal labelling (scale bar: 20 μm). (**J**) Schematic showing the injection strategy of an adeno-associated viral vector (AAV) for Cre-dependent expression of a membrane-bound form of GFP and a synaptophysin–mRuby conjugate in L5a of the MEC in the *Rbp4-Cre* mice. The black outline in the horizontal plane roughly corresponds to the dorso-ventral position of the slice shown in (**K**). (**K**) Immunostained brain section in the horizontal plane showing the distribution of axons (green) and synaptic terminals (red) in the hippocampus as well as the parasubiculum (PaS) and perirhinal cortex (PRh). Injection site in the MEC is distinguishable by the strong fluorescent labelling of L5a cell bodies (scale bar: 400 μm). (**L**) High-magnification image of the injection site in the MEC showing cell membrane and synaptic labelling specifically in L5a (scale bar: 20 μm). (**M**) High-magnification image of the subiculum showing axonal and synaptic labelling (scale bar: 20 μm). High-magnification images of the pyramidal cell layer in distal (**N**) and proximal (**O**) CA1 showing axonal and synaptic labelling (scale bar: 20 μm).

The online version of this article includes the following figure supplement(s) for figure 2:

**Figure supplement 1.** Injection site images and additional examples of axon labelling in CA1.

hippocampus, implying that telencephalic outputs from MEC are copied back to the hippocampus. To distinguish between these possibilities, we used a combinatorial viral labelling strategy. We restricted reporter gene expression to subpopulations of L5a cells that project to specific targets by injecting a retrograde AAV expressing Cre recombinase in either the retrosplenial cortex (RSC) or nucleus accumbens (NucAcb), and a Cre-dependent fluorescent reporter virus into the MEC (*Figure 3A, B* and *Figure 3—figure supplement 1*). With this approach we detected in CA1 fluorescently labelled axons originating from both RSC- and NucAcb-projecting L5a neurons (*Figure 3C, D*). The axon distribution across layers was similar to when L5a neurons were labelled in bulk using the *Rbp4-Cre* mouse line (*Figure 3E, F*). These results establish the principle that axon projections from L5a of MEC to the hippocampus are collaterals of projections to telencephalic targets. However, it is unclear from these experiments whether this principle applies to projections from L5a to all of its many telencephalic targets.

EC outputs have diverse targets, including the entire cortical mantle as well as parts of basal ganglia and amygdala (*Swanson and Köhler, 1986*; *Sürmeli et al., 2015*). To test whether axon collateralization to the hippocampus is a general feature of all telencephalon-targeting neurons, we used Multiplex Analysis of Projections by Sequencing (MAPseq; *Kebschull et al., 2016*). MAPseq is a high-throughput alternative to single neuron anterograde tracing. In MAPseq, instead of multiple copies of a virus carrying the same genetic material, a virus library is utilized where each virus facilitates expression of a unique 30-nucleotide RNA particle. The vast diversity of unique RNA sequences and relatively small number of neurons infected allows each neuron to be marked with a unique RNA barcode (*Kebschull et al., 2016*). Because the barcodes are actively transported to axon terminals, their detection can be used to establish connectivity to putative target structures; if a neuron projects to multiple target structures then its barcode should be detected in each one.

We injected the MAPseq barcode RNA virus library into the full dorso-ventral extent of the deep MEC (*Figure 4—figure supplement 1A*) and quantified barcode RNA expression in tissue collected from dorsal and ventral MEC (dMEC, vMEC), dorsal and ventral hippocampus (dHip, vHip), isocortex, cerebral nuclei (CNU), olfactory areas and cortical subplate (Olf/Ctxsp) as well as brainstem and spinal cord as negative controls (see Materials and methods for detailed description and controls, *Figure 4A*). The majority of barcodes that were detected in any one of the three target divisions (isocortex, CNU, or Olf/Ctxsp) were also detected in the hippocampus, suggesting that collateral axons to the hippocampus are a common feature of telencephalon-projecting neurons (*Figure 4B*). The results were similar regardless of whether neurons were located in dorsal or ventral MEC (*Figure 4—figure supplement 1B*), suggesting that collateralization does not depend on the neuron's position in the dorso-ventral axis of the MEC. Using barcode counts as a measure of projection strength (*Kebschull*

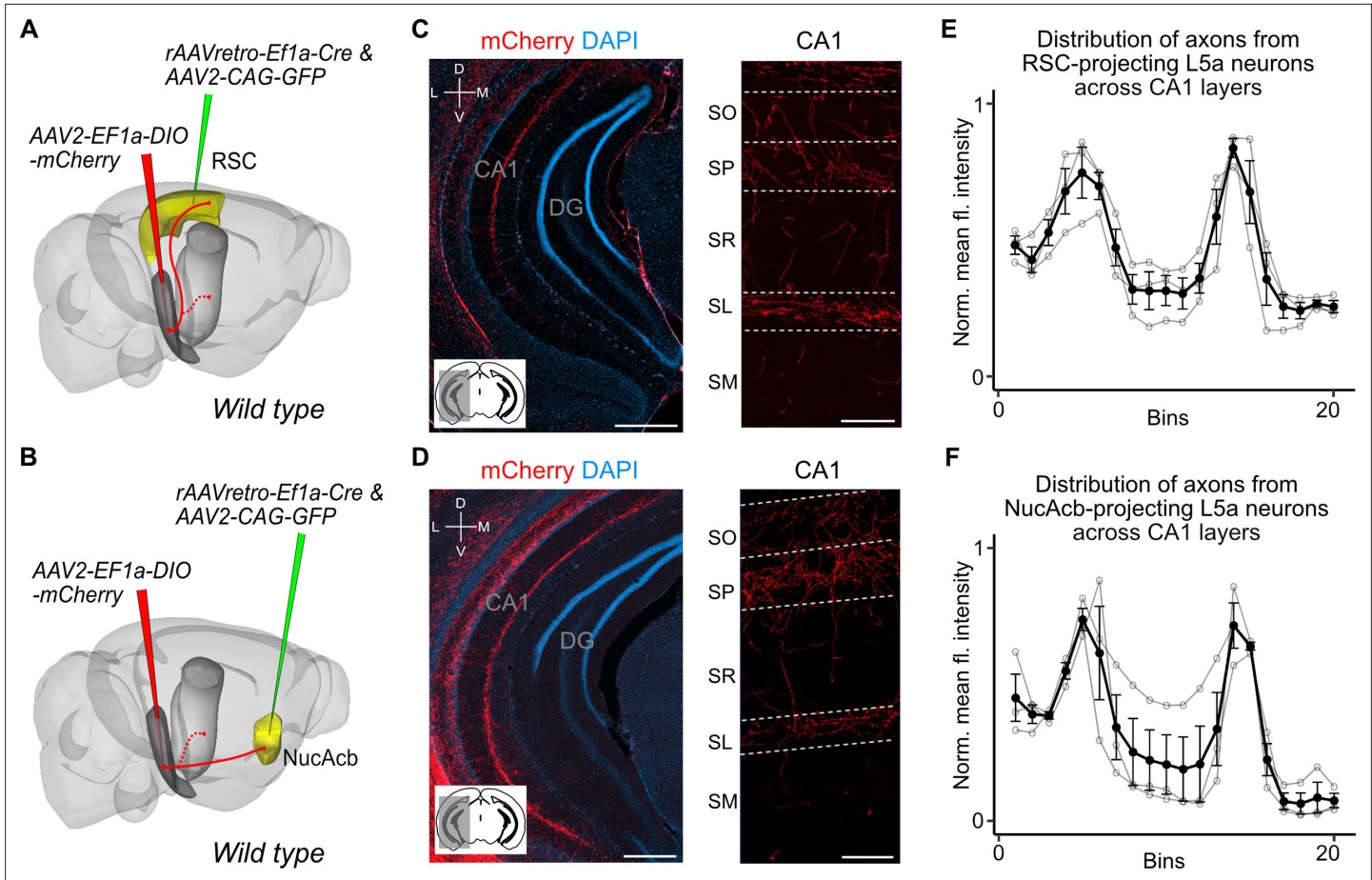

**Figure 3.** Retrosplenial cortex (RSC) and nucleus accumbens (NucAcb) outputs of the medial entorhinal cortex (MEC) are copied to CA1. Experimental strategy. In wild-type mice, a retrogradely transported virus expressing Cre was injected in either the RSC (**A**) or NucAcb (**B**), while a Cre-dependent reporter virus was injected in the deep MEC. Hippocampal axon collaterals of RSC- (**C**) or NucAcb-projecting MEC neurons (**D**; scale bar: 250 μm). Insets: high-magnification images from CA1 showing distribution of axons across CA1 layers (scale bar: 100 μm). Bottom-left schematic indicates the antero-posterior position of the slice. Corresponding injection site images are shown in *Figure 3—figure supplement 1A, B*. (**E, F**) Quantification of mean fluorescence intensity across the layers of CA1 when only RSC- (**E**) or NucAcb-projecting (**F**) MEC neurons were labelled (*n* = 3 mice for each group). Axon topography is similar to when L5a neurons are marked globally in the *Rbp4-Cre* mice (*Figure 2C*).

The online version of this article includes the following figure supplement(s) for figure 3:

**Figure supplement 1.** Combinatorial viral strategy effectively labels L5a cells of the medial entorhinal cortex (MEC) and their cortical projections.

*et al., 2016*) we further found that dorsal MEC neurons preferentially project to dorsal aspects of the hippocampus and ventral MEC neurons to the ventral aspects, indicating that the back-projections to the hippocampus are topographically organized (*Figure 4C*). Together these results reveal general organizing principles by which all projections from MEC to the telencephalon are copied back to the hippocampus.

Which cell types within the hippocampus receive signals from L5a of MEC? While the compartmentalized arrangement of axons from L5a of MEC suggests selective targeting of CA1 layers, axonal topography does not necessarily reflect functional connectivity. Therefore, we targeted viral vectors expressing a channelrhodopsin2–mCherry conjugate to L5a neurons in MEC and tested for connectivity using whole-cell ex vivo patch-clamp recordings from CA1 neurons.

We focused initially on responses of pyramidal cells. Brief light pulses evoked depolarizing subthreshold postsynaptic potentials (PSPs) in 64% of SP pyramidal neurons recorded at their resting membrane potential (*n* = 64 cells, 27 mice, *Figure 5A–C* and *Figure 5—figure supplement 1E*). The response probability of neurons in distal CA1 was not significantly different than neurons in proximal CA1 (*Figure 5D*). Although the vast majority of axons occupy the deep portion of the SP we found no significant difference in the response probability of neurons in deep versus superficial SP (*Figure 5D*).

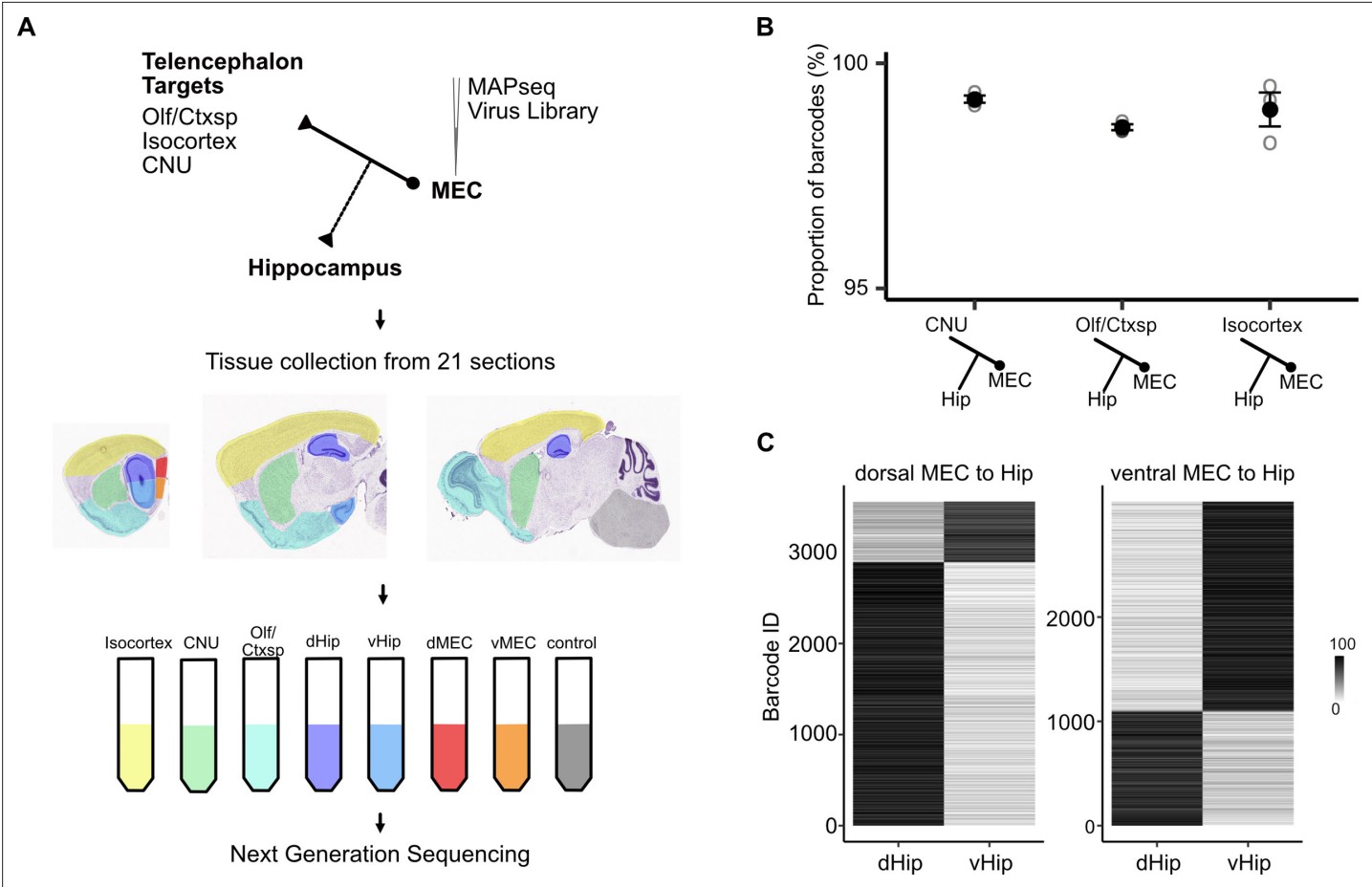

**Figure 4.** Telencephalic outputs are widely copied onto the hippocampus. (**A**) Experimental strategy. A MAPseq virus library was injected into deep medial entorhinal cortex (MEC; *n* = 3 mice). Forty-four hours later, mice were sacrificed and brains were serially sectioned in the sagittal plane at 400 μm thickness. Tissue from five major divisions (isocortex, CNU, Olf/Ctxsp, dHip, and vHip), as well as dorsal and ventral MEC were then dissected and collected in tubes. Brainstem and spinal cord tissue was collected as a negative control. Tissue was further processed for RNA extraction and next generation sequencing. (**B**) Nearly all barcodes detected in CNU, Olf/Ctxsp, and isocortex were also detected in the hippocampus (isocortex: 98.6% ± 0.06%; CNU: 99.2% ± 0.08%; Olf/Ctxsp: 99.0% ± 0.4%; *n* = 3 mice, 1603, 3493, and 1677 barcodes from brains 1, 2, and 3, respectively). (**C**) Dorso-ventral topography of projections revealed by relative barcode counts. For each barcode, counts were normalized to show the relative projection strength of each neuron to dorsal and ventral hippocampus (*n* = 3 mice).

The online version of this article includes the following figure supplement(s) for figure 4:

**Figure supplement 1.** Verification of viral expression and further MAPseq data visualization.

Thus, rather than selecting subtypes of pyramidal neurons axons might target distinct compartments of neurons according to their depth in SP.

The majority of PSPs (20 out of 26) maintained their polarity when the membrane potential was adjusted from rest (Vm = −66.7 ± 0.6 mV) to −50 mV, indicating that they were glutamatergic. In these neurons, EPSPs were maintained when GABA$_A$ receptors were blocked with Gabazine, but were abolished by the AMPA receptor antagonist NBQX, confirming that they are glutamatergic (***Figure 5E***). The responses had short latencies (3.04 ± 0.26 ms, *n* = 20 cells, 15 mice) that were relatively invariant from trial to trial (***Figure 5F***) and were independent of the response's position within a train of stimuli, indicating that they are monosynaptic (***Figure 5—figure supplement 1D***). Consistent with this, responses were sensitive to bath application of tetrodotoxin (TTX) but recoverable after application of 4-aminopyridine (4-AP; ***Figure 5G***; ***Petreanu et al., 2007***). Together, these data demonstrate that projections from L5a of the MEC provide glutamatergic excitatory inputs directly to pyramidal neurons in proximal and distal CA1.

A smaller population of pyramidal neurons showed PSPs with characteristics of indirect inhibitory connections (6 out of 26). These responses either reversed polarity when the cell's membrane

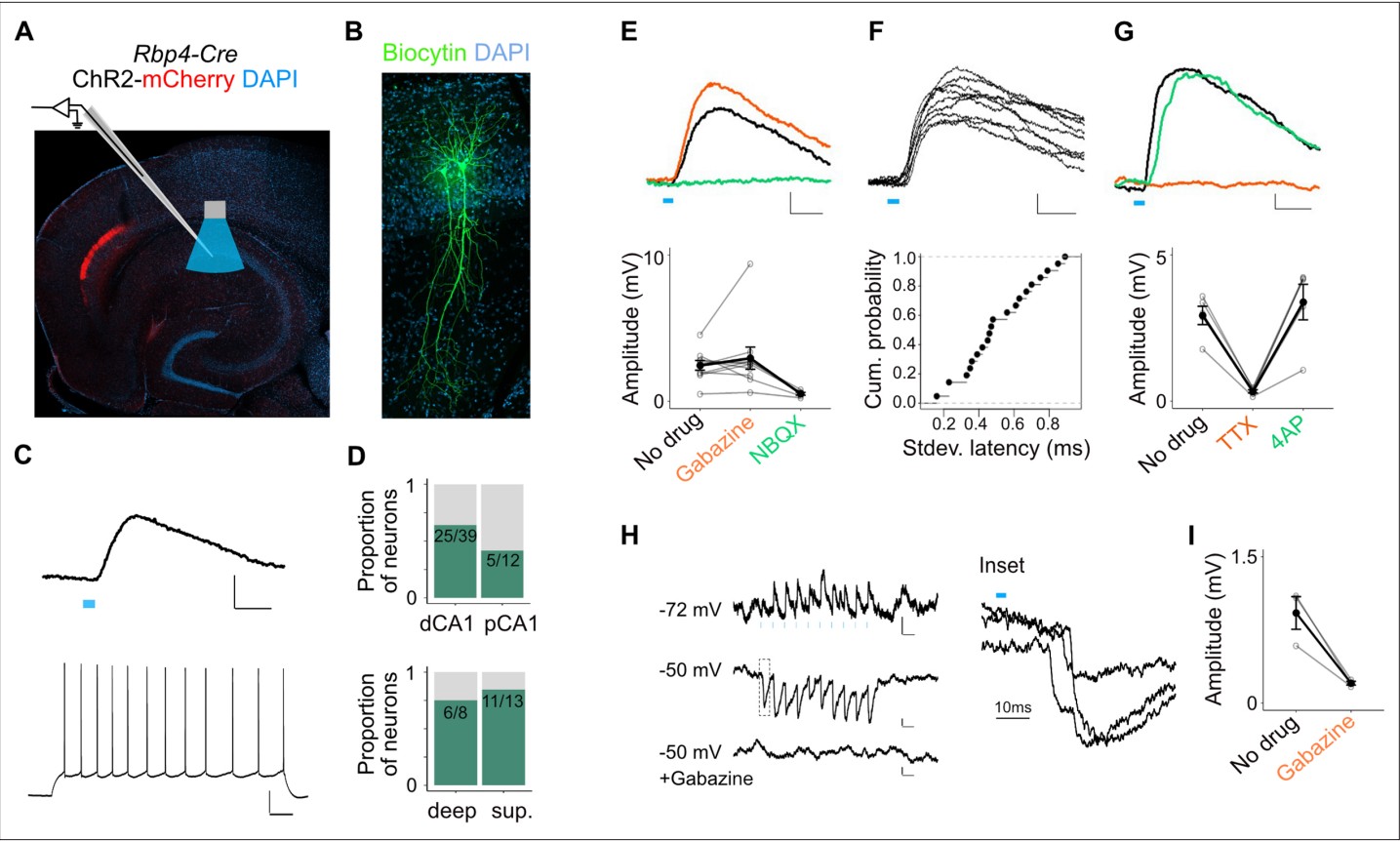

**Figure 5.** L5a neurons of the medial entorhinal cortex (MEC) provide direct excitatory and indirect inhibitory inputs to pyramidal cells in CA1. (**A**) Experimental design showing viral expression, placement of patch-clamp electrode, and light delivery over CA1. (**B**) Examples of biocytin filled pyramidal neurons recorded in CA1. (**C**) An example electrical recording of a pyramidal neuron at rest (−67 mV) showing depolarizing responses upon 3-ms light stimulation (top) (blue line, scale bar: 1 mV, 10 ms). Train of action potentials upon 200 pA step-current injection (bottom; scale bar: 20 mV, 100 ms). (**D**) Proportion of responsive pyramidal neurons located in the distal and proximal halves of CA1 (top). The difference in proportions was not significant ($X$-squared = 1.9071, df = 1, p value = 0.167291, chi-squared test, $n$ = 12 cells in proximal and $n$ = 39 cells in distal CA1). Proportion of responsive pyramidal neurons located in the deep versus superficial locations of CA1 pyramidal cell layer (bottom) ($X$-squared = 0.2969, df = 1, p value = 0.585804, chi-squared test, $n$ = 8 cells in deep SP and $n$ = 13 cells in superficial SP). (**E**) Effects of bath application of Gabazine (orange, $n$ = 10 cells, 9 mice) and NBQX (green, $n$ = 5 cells) on postsynaptic potentials (PSPs) recorded from a pyramidal neuron (scale bar: 0.5 mV, 10 ms) and a summary plot of PSP amplitude measurements for all tested pyramidal neurons. Note that some neurons were only treated with Gabazine which did not cause a significant change in amplitudes (p = 0.97, two-tailed Student's $t$-test, $n$ = 10 cells, 9 mice). (**F**) An example of ten consecutive PSP responses recorded from a single pyramidal neuron illustrates the short and invariant latency of PSPs (scale bar: 0.5 mV, 10 ms) and a cumulative probability plot of standard deviation of latencies for neurons with PSP responses that were >1 mV in amplitude ($n$ = 20 cells). (**G**) Effects of bath application of tetrodotoxin (TTX; orange) and 4-aminopyridine (4-AP; green) on PSPs recorded from a pyramidal neuron (scale bar: 0.5 mV, 10 ms) and a summary plot of changes in PSP amplitudes for all tested pyramidal neurons. TTX application abolished responses ($n$ = 5 cells, 5 mice, p = 0.01, two-tailed Student's $t$-test). (**H, I**) An example inhibitory PSP response recorded from a pyramidal neuron upon 10 Hz light stimulation (blue bars). Response polarity reversed when the neuron's membrane potential was adjusted to −50 mV and was abolished after application of Gabazine (scale bars: 0.2 mV, 100 ms). Inset shows the long latency (>10 ms) of PSP onset indicating polysynaptic connectivity.

The online version of this article includes the following figure supplement(s) for figure 5:

**Figure supplement 1.** Experimental design and properties of responses to optogenetic stimulation of axons originating from L5a of the medial entorhinal cortex (MEC).

potential was held above the chloride reversal potential (***Figure 5H***) or had an early depolarizing and a late hyperpolarizing component (***Figure 5—figure supplement 1B***). Application of Gabazine either completely blocked these PSPs (***Figure 5I***) or revealed a larger excitatory component that was sensitive to application of NBQX (***Figure 5—figure supplement 1B***). Thus, inputs from L5a of the MEC also recruit local interneurons that provide inhibitory input to SP pyramidal neurons.

To identify which interneurons in CA1 were targets of L5a projections, we tested responses of interneurons in all layers. In SP, we distinguished interneurons from principal cells either by classifying

them based on biophysical properties (fast-spiking: $SP_{FS}$ and non-fast spiking: $SP_{NFS}$; *Figure 5—figure supplement 1A* and see Materials and methods), or by recording from fluorescently labelled GABAergic or parvalbumin-expressing (PV+) inhibitory neurons in double transgenic mice (*Figure 6A, B* and *Figure 5—figure supplement 1C*). Stimulation of L5a axons elicited both subthreshold and suprathreshold responses primarily in SP, SR, and SL interneurons (*Figure 6C, D* and *Figure 5—figure supplement 1E*). Depolarizing PSPs were observed in 64% of the $SP_{FS}$ including PV+ interneurons, 46% of the $SP_{NFS}$, 64% of SR, and lower proportions of recorded neurons in SL (31%), SM (23%), and SO (16%). Except for interneurons in SO, PSPs in all layers showed characteristics of monosynaptic connectivity (*Figure 6G, H, I, J* and *Figure 5—figure supplement 1G, H*) and AMPA receptor-mediated glutamatergic synaptic transmission (*Figure 6E, F* and *Figure 5—figure supplement 1F*). Therefore, projections from L5a of MEC recruit CA1 interneurons through monosynaptic inputs driven by excitatory glutamatergic synapses.

## Discussion

Together our results suggest a substantial revision to the idea that the entorhinal deep layers unidirectionally convey hippocampal output messages to the neocortex. Using two lines of evidence, combinatorial viral labelling and MAPseq, we show that L5a axons bifurcate to target both the telencephalon and the hippocampus. Our results imply that nearly all telencephalic output is copied onto the hippocampus regardless of its destination in the telencephalon. There is a topographical register between the dorso-ventral origin of the entorhinal cells and their target zone within the hippocampus. This is similar to the dorso-ventral topography of perforant-path projections of the superficial EC targeting the dentate gyrus, supporting the view that dorso-ventral compartments of the hippocampus operate independently (*Ruth et al., 1982*; *Dolorfo and Amaral, 1998*). Interestingly in the transverse axis, the density of projections has a gradient descending towards proximal CA1 (towards CA2). This is in contrast to previous reports where labelling of MEC projections using traditional tracers revealed dense projections in proximal but not in distal CA1 (*Tamamaki and Nojyo, 1995*; *Naber et al., 2001*). Instead, distal CA1 is thought to be primarily targeted by the lateral EC. The distinct organization of projections originating in L5a of the MEC that we reveal here suggests that they serve functions distinct from the projections targeting CA1 from superficial layers instead of merely supplementing them (*Witter et al., 2017*). This also opens a possibility for the lateral and medial EC input streams to converge in CA1.

Although cells in deep entorhinal layers have been previously observed when retrograde dyes were placed in the CA1-subiculum border (*Köhler, 1985*; *Witter and Amaral, 1991*), the tools we introduced in this study made it possible to investigate the layer origin of these projections and their targets within CA1 with cell-type-specific anterograde labelling. Our observation that L5b does not participate in this pathway reinforces the distinct input–output organization of the deep MEC. While L5b is a point of integration for local and neocortical input with hippocampal output and distributes its signals locally within MEC, based on our anatomical observations L5a appears to be suited to coordinate the wider hippocampal–neocortical loop by providing a copy of entorhinal output back to CA1.

### Ideas and speculation

Our findings raise new questions about the functional roles of outputs from the EC, and about the nature of interactions between the EC and the hippocampus. Telencephalic projections of L5a cells have been shown to be necessary for long-term memory formation and retrieval (*Kitamura et al., 2017*), likely by distributing compressed hippocampal outputs to the neocortex (*McClelland et al., 1995*). Returning a copy of these outputs back to CA1 might create a self-reinforcing or a self-correcting loop. They might also provide a gating signal to modulate the impact in CA1 of inputs from CA3 (*Dudman et al., 2007*) and/or neocortical return inputs via L3 of EC. It is also conceivable that they influence mechanisms of learning and memory by affecting rhythmic network activity through the inputs on the pyramidal cells and interneuron populations that we described here (*Colgin, 2016*; *Ognjanovski et al., 2017*; *Xia et al., 2017*). While our study brings new anatomical insight, more experiments are required to reveal the function of this pathway and the role it plays in spatial cognition, learning, and memory.

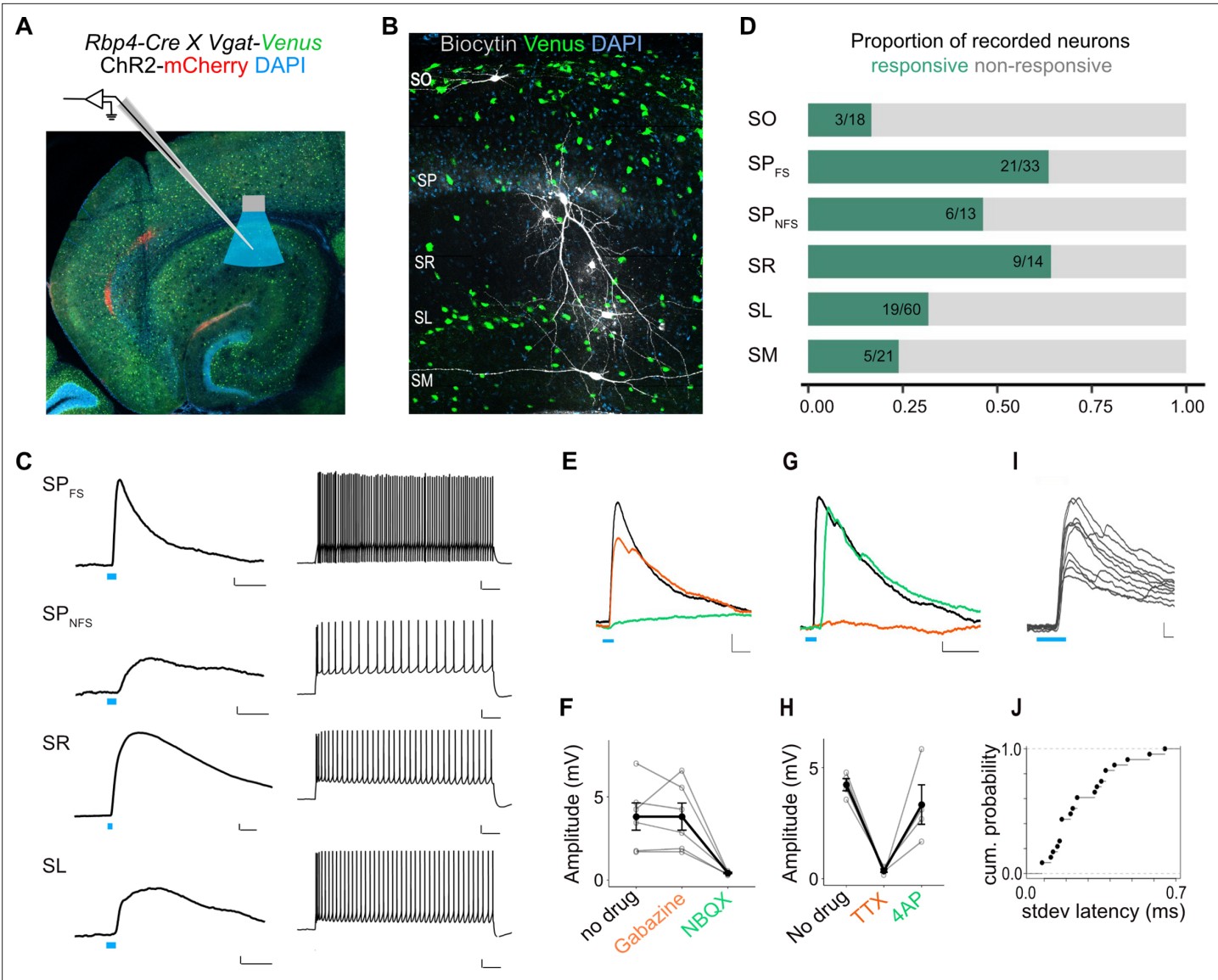

**Figure 6.** L5a neurons of the medial entorhinal cortex (MEC) provide direct excitatory inputs to interneurons in CA1. (**A**) Experimental design showing viral expression, patch pipette, and light stimulation. Interneurons are marked by the expression of venus fluorescent protein driven by the vesicular GABA transporter (*Vgat*) gene's promoter. (**B**) Biocytin filled neurons in the *Rbp4-Cre X Vgat-Venus* double transgenic mouse line. Fluorescent labelling facilitated patch-clamp recordings made from GABAergic interneurons and distinguishing SR, SL, and SM layers. (**C**) Representative examples of depolarizing responses recorded at resting membrane potential following 3ms blue light stimulation (blue line) of L5a axons in CA1 (left) (scale bar: 1 mV, 10 ms). Neurons in SO and SM were typically not responsive (<30%). Responses from SR neurons were on average larger than the responses from neurons in other layers (scale bar: 1 mV, 10 ms). Neurons' spiking response to injecting 200 pA current (right) (scale bar: 10 mV, 100 ms). (**D**) Proportion of responsive interneurons in all layers of CA1 recorded at resting membrane potential. Green highlighted segment corresponds to the proportion of cells with depolarizing membrane potentials; grey highlighted segment corresponds to neurons with no change in their membrane potential. SO: $n$ = 18 cells, 10 mice; SP$_{FSint}$: $n$ = 33 cells, 8 *Rbp4-Cre X Pvalb-Flp* mice, 5 *Rbp4-Cre X Vgat-Venus* mice, 8 *Rbp4-Cre* mice; SP$_{NSFint}$: $n$ = 13 cells, 11 *Rbp4-Cre* mice, 1 *Rbp4-Cre X Vgat-Venus* mouse; SR: $n$ = 14 cells, 6 mice; SL: $n$ = 60 cells, 15 mice; SM: $n$ = 21 cells, 10 mice. (**E, F**) Effects of bath application of Gabazine (orange) and NBQX (green) on postsynaptic potentials (PSPs) recorded from a fast-spiking pyramidal layer interneuron (left) (scale bar: 1 mV, 5 ms). Summary quantification of PSP amplitudes for multiple cells (right). The PSP amplitudes were largely unaffected by application of Gabazine (SP$_{int}$: $n$ = 6 cells, p = 0.99 two-tailed Student's *t*-test) but were largely blocked by NBQX (SP$_{int}$: $n$ = 5 cells, p = 0.03, two-tailed Student's *t*-test) indicating AMPA receptor-mediated glutamatergic synaptic transmission. (**G, H**) Effects of bath application of tetrodotoxin (TTX; orange) and 4-aminopyridine (4-AP; green) on the response amplitude PSPs recorded from pyramidal layer interneurons. Example traces from a fast-spiking pyramidal layer interneuron (scale bar: 1 mV, 5 ms). Summary plots of response amplitude measurements from multiple recordings (SP$_{int}$: $n$ = 4 cells). (**I, J**) An example of ten consecutive PSP responses recorded from a single fast-spiking interneuron in SP illustrates the short and invariant latency of PSPs. Cumulative probability plots of standard deviation of latencies for neurons with PSP responses that were >1 mV in amplitude (mean latency SP: 2.08 ± 0.08 ms, $n$ = 21 cells).

## Materials and methods

### Mice

All animal experiments were approved by the University of Edinburgh animal welfare committee and were performed under a UK Home Office project license. The *Rbp4-Cre* mouse line was generated by GenSat and obtained from MMRRC (*Tg(Rbp4-cre)KL100Gsat/Mmucd*). The *Vgat-Venus* mouse line was generated by Dr. Atsushi Miyawaki at RIKEN, Wako, Japan (*Wang et al., 2009*). The *Pvalb-Flp* mouse line was obtained from Jackson Laboratories (*B6.Cg-Pvalb-tm4.1(FlPo)Hze/J; Jax 022730*). Wild-type mice were obtained from Charles River Laboratories C57Bl6J stock. Double transgenics were generated by crossing the *Rbp4-Cre* line to the *Vgat-Venus* or the *Pvalb-Flp* lines. *Rbp4-Cre* and *Pvalb-Flp* lines were maintained as heterozygous and all mice were on C57Bl/6 background.

### Viral constructs and injection strategy

Eight- to fourteen-week-old male and female mice were used in all experiments. For targeting deep MEC a craniotomy was made 3.4–3.65 mm lateral to the bregma (*X*) between the transverse sinus and lambdoid suture (*Y* = 4.5–5 mm caudal to bregma). The injection pipette was at a 9° angle towards the caudal end of the brain. 100 nl of virus was slowly released at four *Z*-depths 3.0, 2.8, 2.6, and 2.4 mm from the surface of the brain. Three minutes after delivering the virus the injection needle was retracted to the next injection depth until the most dorsal location where 10 min past before the needle was fully retracted. For anterograde tracing of L5a axons and terminals *AAV-EF1a-DIO-mCherry* (serotypes 2 and 5, titre $5.3 \times 10^{12}$, lot #AV4375H, K., Deisseroth, UNC Vector Core), *AAV-CAG-FLEX-GFP* (titre $3.7 \times 10^{12}$, lot #AV4530B, E. Boyden, UNC Vector Core), and AAV-hSyn-FLEx-mGFP-2A-Synaptophysin-mRuby (Addgene:71760) were used.

For injections in CA1 a craniotomy was made 3.5 mm lateral and 3.30 mm caudal to the bregma in C57BL6J mice. 100 nl virus was delivered at 3.0, 2.8, 2.6, 2.4, and 2.2 mm from the surface. For retrograde labelling of hippocampus-projecting EC neurons, 100 nl of a *rAAV2retro-CAG-GFP* (titre $3.5 \times 10^{12}$, lot #AV7493D, E. Boyden, UNC Vector Core) or *rAAV2retro-hSyn-mCherry* (E. Boyden, UNC Vector Core) was injected at each depth.

To assess whether telencephalic projection neurons in L5a of MEC co-express Etv1 (*Figure 1—figure supplement 1D*), fluorophore-coupled Cholera toxin beta subunit (CTB-Alexa 488 and CTB-Alexa 555) was injected at various sites in wild-type mice as previously described (*Sürmeli et al., 2015*) and co-expression of Etv1 and Alexa dyes were examined in confocal images.

To achieve Cre expression in target-specific subpopulation of L5a neurons in the MEC (*Figure 3*), a *rAAVretro-EF1a-Cre-WPRE* (viral construct was a gift from Karl Deisseroth, titre $2.4 \times 10^{12}$) was made using an AAV-retro helper plasmid (Addgene plasmid ID 81070) as described previously (*McClure et al., 2011*). A cocktail of *rAAVretro-EF1a-Cre-WPRE* and *AAV-CAG-GFP* (titre $4.3 \times 10^{12}$, lot #AV6951B, Boyden, UNC Vector Core) viruses were injected either in NucAcb (*X*: +1.0 mm; *Y*: +1.2 mm; *Z*: −3.8, −4.0 mm) or RSC (*X*: +0.4 mm; *Y*: −2.8, −3.2 mm; *Z*: −0.9 mm) in wild-type mice. A Cre-inducible reporter virus was also injected in the MEC as described above. GFP expression driven by the *CAG* promoter at the target site was used for verification of injection location.

To achieve expression of channelrhodopsin-2 in L5a axons, we injected *AAV2-EF1a-DIO-hChR2(H134R)-mCherry-WPRE-pA* (titre $5.1 \times 10^{12}$, lot #AV4319J, Deisseroth) in the MEC of the *Rbp4-Cre* or the *Rbp4-Cre X Vgat-Venus* or the *Rbp4-Cre X Pvalb-FLP* double transgenic mice. Additionally, in order to fluorescently label PV-expressing interneurons in CA1 a Flp recombinase-dependent AAV vector was constructed in house and injected in the hippocampus (*X*: +3.0 mm; *Y*: −3.5 mm; *Z*: −2.0, −3.0 mm, 200 nl virus was injected at each *Z*-depth). To produce the virus, the viral construct pAM-FLEX-GFP (*Murray et al., 2011*) (a gift from Peer Wulff) was used as the viral backbone. The eGFP, loxP, and lox2272 sites were removed and replaced with a synthesized cassette (GenScript) containing a multiple cloning site flanked by two sets of heterotypic (FRT and F3), antiparallel FRT sites to produce pAM FLPX. For pAM FLPX eGFP and pAM FLPX mCherry, the fluorescent proteins were amplified from pAM FLEX eGFP and pmCherry-C1 (Clontech), respectively, with EcoRI and SacI overhangs and cloned into pAM FLPX. AAV preps were generated in house as previously described (*McClure et al., 2011*) and were titred by qPCR following expression in HEK cells (>$10^{12}$ genome copies [GC]/ml).

## Tissue processing and immunohistochemistry

Three to four weeks after the virus injection surgery, mice were transcardially perfused with 4% paraformaldehyde as previously described (*Sürmeli et al., 2015*). Brains were sectioned at 60 μm using a cryostat (Leica CM3050 S) either coronally or horizontally and sections were stored in phosphate-buffered saline (PBS) at 4°C. All subsequent incubation and washing steps were carried out at 4°C. Prior to antibody staining, sections were blocked with either 5% normal goat serum (NGS; Abcam: ab13970) or 2% bovine serum albumin (BSA; Sigma: A9418) in 0.3% PBST (PBS with 0.3% Triton X-100) for 2 hr. Sections were then transferred to primary antibody solutions made with either 5% NGS or 2% BSA in 0.3% PBST and incubated overnight. After three washes, each 20 min, in 0.3% PBST, sections were transferred to secondary antibody solutions made with 0.3% PBST and, if required, NeuroTrace 640/660 (1:800; Life Technologies: N21483) and incubated overnight. After three washes, sections were incubated in DAPI (1:2000; Sigma-Aldrich: D9542) solution made in PBS for 20 min at room temperature, where required, and then mounted on microscope slides using Mowiol 4-88 (Aldrich: 81381). Slides were covered with glass coverslips and left to dry overnight at 4°C in the dark. The following primary antibodies were used: rat anti-mCherry (1:2000; Thermo Fisher: M11217), chicken anti-GFP (1:10,000; Abcam: ab13970), rabbit anti-Etv1 (*Arber et al., 2000*) , and chicken anti-NeuN (1:1000; Sigma-Aldrich: ABN91). The following secondary antibodies from Thermo Fisher Scientific were used at a concentration of 1:800: goat anti-rabbit Alexa Fluor 488 (A11034), goat anti-chicken Alexa Fluor 488 (A11039), goat anti-rat Alexa Fluor 555 (A21434), and goat anti-rabbit Alexa Fluor 647 (A21244).

## Cell counting

Following sectioning, every fourth brain section was imaged as a Z-stack (1 μm step size) using either the Zeiss LSM 800 (Zen, v2.6.76) or Leica SP8 confocal microscope (Leica Application Suite X, v3.5.6.21594) at ×20 magnification. Regions of interest were drawn around L5a and cell counting was carried out either manually using Fiji software or using cell-counting tools in Vision4D (Arivis, v3.2.0).

The boundaries for L5a were determined using either a DAPI or Neurotrace counterstain. The border between medial and lateral divisions of EC was determined in each section using layer 2 as a guide – in lateral EC, layer 2 is separated into two distinct layers, while this separation is not seen in medial EC (*Cappaert et al., 2015*).

## Quantification of the distribution of fluorescent signal

### Quantification of fluorescence across the radial axis of the CA1

The distribution of axons of L5a neurons in the radial axis of the hippocampal CA1 was quantified in slide-scanner images (*Figure 2*) or confocal images (*Figure 3*) using Fiji software. Coronal brain sections between 3.28 and 3.58 mm posterior to Bregma, which contain distal CA1, were selected for analysis. Two sections from each brain in which the fluorescent labelling was representative of all the sections were used. DAPI was used as a counterstain. In each section, a 400-μm wide rectangular ROI was drawn across the radial axes, covering all layers of a randomly selected region in intermediate CA1 (*Figure 2B*). This ROI was then divided into 20 equal bins and the mean fluorescence intensity was calculated for each bin. The mean intensity values were then normalized in each brain by scaling the values such that the highest value is 1.

### Quantification of fluorescence in the transverse axis of the CA1

A Zeiss Axioscan slidescanner was used to image every second brain section at ×10 magnification. The projection strengths of MEC L5a neurons to proximal and distal halves of CA1 were quantified in horizontal sections located at depths between 2.56 and 4.12 mm from the surface of the brain, with injection sites located across medial, mid and lateral MEC (15–18 brain sections per mouse, *n* = 5 mice). GFP and mCherry fluorescence signals were amplified with immunostaining as described previously. The borders of CA1 were drawn on brain section images using a customisable digital microscopy analysis platform (Visiopharm), and the proximo-distal border was defined as the border equidistant from the proximal and distal ends of CA1. The mean fluorescence density, defined as the total number of pixels above a set threshold in an area divided by the total number of pixels in the area, was measured in proximal and distal CA1. The threshold was determined manually by ensuring that only pixels representing axons were detected as signal. A median unsharp background correction

was used to remove background noise from axons outside of the focus plane of the image. The mean fluorescence density values were then normalized within each brain by scaling the values such that the total fluorescence density value (proximal + distal) in each brain is equal to 1.

The spread of viral expression in MEC was assessed from the fluorescent signal in all sections for each brain. Brains in which labelled neurons were found outside L5 were excluded; this was a result of occasional and sparse labelling in L2 or L3 or the parasubiculum in the *Rbp4-Cre* line.

### MAPseq

5 C57Bl6J adult male and female mice were injected in the deep MEC with the MAPseq Sindbis viral barcode library ($3 \times 10^{10}$ GC/ml) provided by the MAPseq facility (Cold Spring Harbor Laboratories). To cover the whole mediolateral and dorso-ventral extent of the MEC virus was injected in two locations: 3.4 and 3.6 mm lateral to bregma. 100 nl virus was delivered to 3.0, 2.8, 2.6, and 2.4 mm below the surface of the brain. After 44 hr, mice were sacrificed, brains were extracted and immediately immersed in oxygenated cold artificial cerebrospinal fluid (ACSF prepared in ultrapure water). All surgical tools and surfaces were treated with RNaseZAP (Invitrogen) prior to the start of the experiment and in-between samples. 400 µm fresh sagittal brain sections were cut using a vibratome (*Pastoll et al., 2012*). Immediately after this, GFP expression in the deep EC was verified under a fluorescent microscope and tissue was dissected in cold oxygenated ACSF using microdissection blades on sylgard covered Petri dishes which were kept on ice. Each slice was dissected on a previously unused surface of the plate with fresh ACSF and a dedicated microblade was used for the dissection of each brain division to prevent contamination. Tissue pieces were collected into bead containing (Qiagen 69989) collection tubes (Qiagen 19560) on dry ice. After tissue collection, 400 µl Trizol (Thermo Fisher Scientific #15596026) was added to each collection tube. The procedure was repeated for all five brains. Three brains with the largest coverage of MEC deep layers were selected to proceed to the next steps. Tissue was stored at −80° before it was shipped on dry ice to CSHL MAPseq core facility for further processing. RNA extraction, production of double-stranded cDNA, library preparation, and Illumina sequencing and preprocessing of sequencing data were performed by the MAPseq core facility as described in *Kebschull et al., 2016*.

For the tissue dissections, identification of brain areas was done by using Allen Brain Reference Atlas (https://mouse.brain-map.org/). The isocortex division included the somatosensory, motor, visual, posterior parietal, anterior cingulate, and retrosplenial areas combined. The CNU division was restricted to striatum. The olfactory/cortical subplate division (Olf/Ctxsp) was a combination of olfactory areas and cortical subplate including amygdalar nuclei. The remaining two divisions were dorsal (dHip) and ventral hippocampus (vHip) including subiculum. Some brain areas were excluded from the study because of the difficulty in dissecting or identifying brain areas in the sagittal plane. All sections >3.7 mm lateral to bregma are not annotated in Allen Brain Reference Atlas and were excluded. Therefore, neocortical areas in the most lateral sections such as perirhinal cortex, ectorhinal cortex, and temporal association areas were not included in the study. The claustrum and adjacent neocortical areas (visceral, agranular insular, gustatory) were excluded as it was not possible to separate these areas precisely to prevent contamination between the assigned divisions. Since borders between the brain divisions CNU and OLF/Ctxsp were not always clear, dissections avoided these areas hence the brain areas in these divisions are partially included. White matter between the hippocampus and the neocortex carrying axon tracts were also excluded. Brainstem and cervical spinal cord tissue were used as control. When L5a neurons in the MEC were labelled using strategies explained in *Figure 2*, no axonal projections to these areas were observed (unpublished experiments). Consistent with this, a low number of barcodes were identified in the negative control samples (on average 1.0% ± 0.3% of barcodes had counts in control areas, *n* = 3).

A limitation of the barcode-based single-cell tracing method comes from the possibility of multiple neurons being represented by the same barcode. This is prevented by adjusting the barcode diversity and the size of the target population as explained in detail in *Kebschull et al., 2016*. In order to assess the expected fraction of uniquely labelled neurons in our study we counted the total number of neurons in L5a of the MEC. A total of 2314 ± 131 (*n* = 2 mice) NeuN-labelled neurons were counted. Using the formula $F = (1 - (1/N))^{k-1}$ where $N$ is the barcode diversity ($2 \times 10^6$) (*Kebschull et al., 2016*) and $k$ is the number of infected neurons the predicted ratio of uniquely labelled neurons is 99.9%. Multiple representation of neurons due to a neuron being infected by several different barcode

RNA-expressing viral particles was not corrected for since the projection patterns are not affected by overrepresentation of neurons (*Kebschull et al., 2016*).

## MAPseq data analysis

Barcode counts were first normalized in each area by the relative number of spike-in RNAs for each sample. Orphan barcodes, barcodes which did not have counts in the injection site (dMEC or vMEC) were removed. We then calculated the 90th percentile of the barcode counts in our negative controls and based on this set all barcode counts of 1–0. A small number of barcodes had a higher count in any target area compared to the injection site which might be a result of incomplete dissection of the injection site or viral expression in areas that the virus spilled into as we retracted the pipette. These barcodes were removed. Since our goal was to find whether neurons projecting to the telencephalon also project to the hippocampus we removed the barcodes that had no counts in any of the telencephalic target areas. For the same reason, barcodes that were detected only in the hippocampus were also removed. Finally, we excluded all barcodes with counts of less than 400 in the injection site to minimize the possibility of incomplete transport of RNA barcodes to axons in target areas due to weak expression at the cell bodies or low counts due to PCR or polymerase errors. Barcodes with low counts in all target areas (<10) were also excluded to account for potential false positives.

To quantify the proportion of barcodes that were present in both a target division (*Figure 4B*; isocortex, CNU, or Olf/Ctxsp) and the hippocampus, the following formula was used: (total number of barcodes with counts detected in both hippocampus and target division/total number of barcodes with counts detected in target division). To classify the barcodes as originating from dMEC vs vMEC, the barcodes were divided into two groups – 'dorsal MEC', where counts in dMEC were higher than in vMEC, and vice versa for 'ventral MEC'.

For a visual display of the relative barcode counts in dHip and vHip of barcodes originating from 'dorsal MEC' or 'ventral MEC', we generated heat maps showing normalized projection strengths between dHip and vHip for all barcodes (*Figure 4C*). The normalized projection strengths were calculated by the formula: projection strength$_{dHip}$ = (counts$_{dHip}$)/(counts$_{dHip+vHip}$) × 100, and likewise for vHip. Barcodes were sorted by maximum projection site.

## Ex vivo electrophysiology

### Slice preparation

Slice preparation and subsequent data acquisition were done as previously described (*Sürmeli et al., 2015*). Three to four weeks after the injection of the viral vectors ex vivo brain slices were prepared. Following decapitation, brains were immersed for 2 min in 4°C ACSF of the following composition (mM): 86 NaCl, 1.2 NaH$_2$PO$_4$, 2.5 KCl, 25 NaHCO$_3$, 25 glucose, 75 sucrose, 0.5 CaCl$_2$, 7 MgCl$_2$, bubbled with 95% O$_2$/5% CO$_2$. They were then sectioned horizontally (400 µm) using a vibratome (Leica VT1200) ACSF. Tissue was collected and maintained in extracellular solution of the following composition (mM): 124 NaCl, 1.2 NaH$_2$PO$_4$, 2.5 KCl, 25 NaHCO$_3$, 20 glucose, 2 CaCl$_2$, and 1 MgCl$_2$, continuously supplied with 95% O$_2$/5% CO$_2$. Slices were allowed to rest for 15 min at 35°C followed by a minimum of 30 min recovery time at room temperature before the start of the experiment.

### Electrophysiological recordings

Whole-cell patch-clamp recordings were made in pyramidal neurons and interneurons in all layers of hippocampal CA1. Typically, two to three slices from the intermediate hippocampus were used where morphologies of the pyramidal cells and interneurons were confirmed to be intact with post hoc staining and imaging of biocytin filled neurons. Data were collected using AxoGraph (v1.7.6) software.

Pipettes with 4–6 MΩ resistance were pulled from borosilicate glass (Sutter Instruments) and filled with an intracellular solution of following composition (mM): 130 K gluconate, 10 KCl, 10 HEPES, 2 MgCl$_2$, 0.1 EGTA, 2 Na$_2$ATP, 0.3 Na$_2$GTP, 10 NaPhosphocreatine, 5.4 Biocytin. The intracellular solution was adjusted to a pH of 7.2–7.4 with KOH and osmolarity 290–300 mOsm. All recordings were made in current clamp mode with pipette capacitance neutralization and bridge balance applied. Subthreshold membrane properties were measured from the changes in membrane potential upon depolarizing and hyperpolarizing current injections (typically −40 to +40 pA, in 20 pA). Rheobase was established from responses to a depolarizing current ramp (50 pA/s, maximum current 400 pA).

For optogenetic stimulation of ChR2, an LED of wavelength 470 nm (ThorLabs) was attached to the epifluorescence port of the microscope. Where necessary, the irradiance of the LED (max 9 mW) was controlled by voltage commands. Pharmacological tests were done by bath application of the following reagents with the indicated final concentrations in standard extracellular solution: Gabazine (10 µM, Hello Bio, Cat. No. HB0901), NBQX (10 µM, Tocris, Cat. No. 0373), D-AP5 (50 µM, Tocris, Cat. No. 0106), 4-AP (200 µM, Tocris, Cat. No. 0940), and TTX (1 µM, Hello Bio). To allow for morphological reconstructions cells were filled with Biocytin (5.4 mM in intracellular solution, Sigma, Cat. No. B4261) during electrophysiological recordings.

## Analysis of electrophysiological recordings

Electrophysiological properties were analysed using built-in and custom routines in IGORpro8 (WaveMetrics) and Matlab (MathWorks). All basic properties were established from the I–V protocol described above. Input resistance was determined from the largest depolarizing current step injected. Sag ratio was calculated as $V_{steady\ state}/V_{min}$ from the largest hyperpolarizing current step. Rheobase and action potential (AP) threshold were determined from the first AP during injection of steadily increasing current in a ramp. From the same protocol, after-hyperpolarization (AHP) was calculated as the difference between AP threshold and the most negative peak of the hyperpolarization following the first spike. Half-width was measured as the width of the AP at half its maximum spike amplitude. Firing frequency was measured from a 1-s current injection of 200 pA. Maximum firing frequency was measured between the first two APs, and base frequency between the last two APs in a train.

For optogenetic stimulations, responsiveness was confirmed for each cell by a significant difference between the detected peak of change in membrane potential after light stimulation and the average baseline using a two-tailed, type one Student's *t*-test. The PSP amplitude and latency were measured using the Neuromatic toolbox in IGORPro8.

Cell types were determined based on biophysical properties extracted from electrophysiology data. Pyramidal cells were identified from a max firing frequency of <50 Hz, AHP <10 mV, AP half-width ≥0.7 ms, and sag ≤0.9, with descending hierarchical order. Fast spiking interneurons were categorized from a max firing frequency ≥100 Hz, AHP ≥12 mV, AP half-width <0.5 ms, and sag >0.9. Non-fast spiking interneurons were classified from a max firing frequency 50–100 Hz, AHP ≥12 mV, AP half-width 0.5–0.9 ms, sag ratio >0.9. and max firing frequency 50–100 Hz.

Neurons with a resting membrane potential less negative than −50 mV or a bridge balance higher than 40 MΩ were excluded from the analysis, as were neurons that did not fit within a class from above criteria.

The proximo-distal position and sublayer identity of neurons filled with biotin during patch-clamp recordings were determined post hoc on immunostained slices after streptavidin staining. CA1 was divided into two halves by drawing a line equidistant to CA1/CA2 and CA1/Sub border. Only neurons that were clearly located in the proximal and distal halves were included in the analysis. Neurons from the first two cell layers of SP were assigned a superficial, neurons in deeper parts of SP towards SO were assigned a deep sublayer profile. Neurons in intermediate positions were excluded from the analysis since it was not possible to unequivocally determine whether they belong to deep or superficial SP.

## Immunostaining of electrophysiology slices

For identification of cells following recordings, tissue was fixed in 4% paraformaldehyde overnight at 4°C. Slices were washed with PBS three times, 20 min each, and incubated in Streptavidin-Alexa 488 (S11223) or Alexa 647 (1:500, S32357, Invitrogen) and DAPI (1:1000, in 0.3% PBST) overnight at 4°C. Slices were washed in 0.3% PBST four times and mounted on glass slides with Mowiol 4-88. Mcherry fluorescence in the MEC was examined with confocal imaging and those animals where virus injection missed MEC L5 were excluded.

For staining of the interneuron marker Parvalbumin (PV), slices were prepared as described above, then incubated in primary antibody solution containing mouse anti-PV (1:1000, PV 235, Swant) and 5% NGS in 0.3% PBST for 48 hr at 4°C. Slices were then washed and incubated in secondary antibody solution with Alexa-conjugated Streptavidin and DAPI prepared in 0.3% PBST and mounted on glass slides after overnight incubation.

A Zeiss LSM800 microscope was used for image acquisition of the slices. Pinhole was set to 1 Airy Unit. Objectives used include ×10 (air), ×20 (air), and ×40 (oil) to image the hippocampal formation, morphology of biocytin filled cells, and immunolabelling of interneurons, respectively.

### Image processing and generation

Experimental design sketches were generated using Brainrender (*Claudi et al., 2021*) or Affinity Designer. Where necessary images were cropped and size-adjusted using Photoshop, Pixelmator, and ImageJ softwares.

### Statistics

Statistical tests (Student's *t*-test, chi-squared test, Wilcoxon signed rank test) were performed using R (https://www.r-project.org) 3.6.0, 4.0.0, and 4.0.3. Normality was tested for using both the Shapiro–Wilk test and *Q–Q* plots. All data are presented in the format of mean ± standard error of the mean unless otherwise stated. A p value <0.05 was regarded as significant.

## Acknowledgements

We thank Matt Nolan and Nolan Lab members for discussions and sharing reagents and equipment. We thank Justus Kebschull for sharing his insight in MAPseq technique, Maria Doitsidou for sharing her equipment and lab space. We also thank Matt Nolan, Klara Gerlei, Brianna Vandrey, Christina Brown, and Sürmeli lab members for critical reading of the manuscript and Gamze Şener for help with early experiments. CM was supported by a BBSRC project grant BB/M025454/1. This research was supported by a Royal Society and Wellcome Trust Sir Henry Dale fellowship 211236/Z/18/Z.

## Additional information

### Competing interests

Christina McClure: is affiliated with VectorBuilder Inc. The author has no financial interests to declare. The other authors declare that no competing interests exist.

### Funding

| Funder | Grant reference number | Author |
| --- | --- | --- |
| Wellcome Trust | 211236/Z/18/Z | Gulsen Surmeli |
| Royal Society | 211236/Z/18/Z | Gulsen Surmeli |
| Biotechnology and Biological Sciences Research Council | BB/M025454/1 | Christina McClure |

The funders had no role in study design, data collection, and interpretation, or the decision to submit the work for publication.

### Author contributions

Sau Yee Tsoi, Ella Svahn, Formal analysis, Investigation, Methodology, Visualization, Writing – original draft; Merve Öncül, Formal analysis, Investigation, Writing – original draft; Mark Robertson, Formal analysis, Investigation, Methodology, Visualization; Zuzanna Bogdanowicz, Formal analysis, Investigation, Visualization; Christina McClure, Methodology; Gülşen Sürmeli, Conceptualization, Data curation, Formal analysis, Funding acquisition, Investigation, Methodology, Project administration, Resources, Supervision, Visualization, Writing – original draft, Writing – review and editing

### Author ORCIDs

Gülşen Sürmeli  http://orcid.org/0000-0002-3227-0641

### Ethics

All animal experiments were approved by the University of Edinburgh animal welfare committee and were performed under a UK Home Office project license.

Decision letter and Author response
Decision letter https://doi.org/10.7554/eLife.73162.sa1
Author response https://doi.org/10.7554/eLife.73162.sa2

## Additional files

### Supplementary files
• Transparent reporting form

• Source data 1. MAPseq barcode count matrix. Source data used to generate all plots in *Figure 4*, *Figure 4—figure supplement 1*. Barcode counts were filtered and normalized by spike-in counts as described in methods. Column names are abbreviations of brain area names as explained in figure legends.

### Data availability
MAPseq FASTQ sequences are publicly available via University of Edinburgh's Datashare service (https://datashare.ed.ac.uk/handle/10283/4369). All other source data are available upon request.

The following dataset was generated:

| Author(s) | Year | Dataset title | Dataset URL | Database and Identifier |
|---|---|---|---|---|
| Surmeli G, Tsoi SY | 2022 | MAPseq FASTQ sequences and barcode count matrix Tsoi et al., 2022 | https://doi.org/10.7488/ds/3420 | Edinburgh DataShare, 10.7488/ds/3420 |

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
