## [Editor Report]

In this work, the authors combine a variety of novel circuit mapping techniques to characterize a novel projection pathway from layer 5a neurons in the medial entorhinal cortex to region CA1 of the hippocampus. By utilizing cell specific viral labelling techniques, RNA sequencing based projection mapping, and optogenetic-aided in vitro physiology, the authors show evidence that the same neurons in layer 5a of medial entorhinal cortex project to both cortical areas of the telencephalon and the hippocampus. This work raises the possibility that deep layers of the entorhinal cortex coordinate hippocampal-cortical interactions. The manuscript will be of interest to readers in the field of anatomy and hippocampal physiology.

---

## [Decision Letter]

**Decision letter after peer review:**

[Editors’ note: the authors submitted for reconsideration following the decision after peer review. What follows is the decision letter after the first round of review.]

Thank you for submitting the paper "Telencephalic outputs from the medial entorhinal cortex are copied directly to the hippocampus" for consideration by *eLife*. Your article has been reviewed by 3 peer reviewers, and the evaluation has been overseen by a Reviewing Editor and a Senior Editor. The following individual involved in review of your submission has agreed to reveal their identity: Kevin Allen (Reviewer #3).

Comments to the Authors:

We are sorry to say that, after consultation with the reviewers, we have decided that this work will not be considered further for publication by *eLife*. While the reviewers felt the paper presented robust analyses of novel anatomical pathways, without functional evidence for how these pathways interact or support memory processing, the interest may be limited to specialists and not broad enough for the readership of *eLife*. However, if additional experiments were to lead to insight regarding the function of these pathways, *eLife* would consider a new submission of the manuscript. Please note in this case, the manuscript would be treated as anew submission.

*Reviewer #1:*

This manuscript characterizes where deep layers of the medial entorhinal cortex (layer 5a and layer 5b) project to in the brain. Using a variety of circuit mapping techniques (cell-type specific anatomical tracing, high-throughput RNA sequencing based projection mapping and optogenetics aided circuit mapping), the authors find that the same neurons in the layer 5a of the medial entorhinal cortex send projections to both the telencephalon and the hippocampus. They also find that the projections target hippocampal pyramidal cells and interneurons and has a unique topography. While these findings are interesting and suggest that deep layers of the entorhinal cortex may coordinate hippocampal-cortical interactions in memory processing, but this is just speculation based on the anatomical connections.

The manuscript does a great job of characterizing the anatomical projections with a variety of approaches and the results are interesting. However, without functional evidence of how these projections interact with other brain regions to support cognition or memory processing, I think the current version of the manuscript does not quite meet the impact level for publication at *eLife*.

*Reviewer #2:*

In this study, the authors have characterized a novel projection from layer 5a of entorhinal cortex to CA1 of hippocampus. Overall the data are convincing that 5a cells project to CA1. The optogenetic experiments provide strong evidence that the projection to CA1 is glutamatergic and targets pyramidal cells and several classes of interneurons. The authors present some evidence that the same 5a cells send axon collatorals to retrosplenial cortex + CA1 and nucleus accumbens + CA1 - however these data could be strengthened by making it more clear whether only 5a cells were targeted within the MEC. This reviewer does not have the expertise to comment on the MAPseq experiments.

The data characterize the projection well - but lack any kind of functional insight. Without this, it is difficult to place these findings in context. For example, are the projections to different interneuron subtypes different? Are there conditions when input to CA1 would be strong or weak - in other words - what are the dynamics of the layer 5MEC to CA1 pathway? Does silencing this projection effect behavior?

In summary - these data nicely characterize a novel projection from layer 5a of MEC to CA1, but leave open questions as to the function of this projection.

Regarding the axon collateral experiments: In Figure 3- supplement 1, it was unclear to me whether only layer 5a is targeted. Could you label layers on the figure to ensure this? Did you consider doing retrograde injections in CA1 and in RSC/NucAcb to look for double labeled 5a cells?

Regarding the optogenetic aided circuit mapping experiments: These comprise a large set of recordings which are technically challenging. I was hoping, however, that you might go beyond the characterization of connections to give some functional context to this pathway. In the abstract you state – 'Our results suggest that rather than serving as a relay, deep EC may coordinate hippocampal-neocortical interactions in spatial cognition and memory'. I think this statement requires experiments aimed at understanding the dynamics and magnitude of the connection at the very least.

*Reviewer #3:*

Layer 5 neurons of the entorhinal cortex are thought to play a key role in memory consolidation because they receive inputs from the hippocampus and send axons to large parts of the neocortex (Rosov et al., 2020; Witter et al., 2017). Six years ago, it was shown that Layer 5 neurons of the entorhinal cortex can in fact be divided into 2 populations (Layer 5a and 5b) with very different axonal profiles (Surmeli et al., 2015). Neurons in Layer 5a, but not those in layer 5b, project to the neocortex. Based on these findings, Layer 5a neurons appear perfectly suited to influence neocortical areas and contribute to memory consolidation.

The current manuscript by Tsoi et al. reveals an important new twist to our understanding of this Layer 5 output pathway: a large proportion of layer 5a neurons that project to the telencephalon (including the neocortex) also provide excitatory inputs to the hippocampus. The idea that neurons in the deeper layers of the entorhinal cortex send axons to the hippocampus is not new (Köhler, 1985; Witter and Amaral, 1991), but, until now, it was not clear if the population of neurons projecting to the neocortex was also sending axons to the hippocampus.

Tsoi and colleagues make use of two modern tracing strategies to test whether Layer 5a neurons projecting to the telencephalon also project to the hippocampus. First, they inject a retrograde AAV expressing Cre-recombinase in extra-hippocampal areas targetted by Layer 5a neurons. A second AAV expressing a reporter protein in a Cre-dependent manner was also injected in the deep layer of the medial entorhinal cortex. The Cre-dependent reporter protein was observed in the CA1 areas. The second approach used is MAPseq. A MAPseq barcode RNA virus library was injected into the deep layers of the medial entorhinal cortex. With this technique, the majority of infected neurons are expected to express a unique RNA sequence, which will be present both in the cell body the axon terminals. The entorhinal cortex, hippocampus, and other telencephalic structures were then processed to identify the bar codes present in the different target areas of Layer 5a neurons. The majority of barcodes found in the telencephalic structures were also found in the hippocampus, suggesting that a very large proportion of Layer 5a neurons projecting to the hippocampus also project to the hippocampus.

The manuscript has a clear message and the conclusions are generally well supported by the data presented. Two complementary methods are used to show that MEC deep layer neurons projecting to the telencephalon also project to the hippocampus. The reported proportion of Layer 5a neurons having projections towards the telencephalon and the hippocampus is very high, suggesting that this is an important feature defining this cell population.

One limitation of this work is that the functional role of the axon collaterals in the hippocampus is not explored in detail. The main target cells in the CA1 areas have however been identified. In addition, the authors describe a mouse line in which Cre-recombinase in the entorhinal cortex is limited to Layer 5a neurons. These mice will surely prove useful in future studies investigating the role of Layer 5a neurons.

It is not clear whether there are differences in connectivity between Layer 5a and CA1 superficial or deep pyramidal cells.

The MAPSeq technique is relatively new and the percentage of shared barcodes between the hippocampus and the telencephalic areas is very high (approaching 99%). Given these high values, more control data would be beneficial. What is the overlap of barcodes across animals or between telencephalic brain areas? It would be interesting to know whether the layer 5a neurons projecting to the Isocortex also project to the CNU? The results on the control brain area shown in the Methods section (line 433) could be moved to the Results section.

Line 209: Does light stimulation lead to suprathreshold PSP in CA1 pyramidal cells? If so, how confident can we be that the responses observed in interneurons are not due to feedback connectivity between spiking CA1 pyramidal cells and inhibitory neurons? Is the latency sufficiently low to rule out indirect connectivity? It would be beneficial to show the response latency for these responses.

Line 159: The MAPSeq technique is relatively new compared to viral tracing and many readers might not know how it works. I would suggest adding a few sentences explaining this method.

---

## [Author Response]

[Editors’ note: The authors appealed the original decision. What follows is the authors’ response to the first round of review.]

Reviewer #1:The manuscript does a great job of characterizing the anatomical projections with a variety of approaches and the results are interesting. However, without functional evidence of how these projections interact with other brain regions to support cognition or memory processing, I think the current version of the manuscript does not quite meet the impact level for publication at eLife.

We are pleased that all reviewers found our results robust, clear and interesting. The reviewer naturally questions the functional role of this pathway. We believe the characteristics of the pathway that we revealed in this manuscript and the tools we introduce will be the foundation of many experiments targeted to reveal its function. Our findings will also influence new circuit models of learning and memory, a popular area of theoretical neuroscience. Therefore, we firmly believe it is impactful and appropriate for the readership of *eLife* in its current state.

Reviewer #2:Regarding the axon collateral experiments: In Figure 3- supplement 1, it was unclear to me whether only layer 5a is targeted. Could you label layers on the figure to ensure this?

We thank the reviewer for this point. We understand that readers might not be familiar with the appearance of layers of the EC in coronal sections. The curved geometry of the medial entorhinal cortex makes layer organization in caudal sections in the coronal plane unintuitive: L5 cells look like a clump as opposed to a thin layer of neurons that most readers would be familiar with from horizontal or sagittal sections as in Figure 1A-D. Therefore, it is difficult to demarcate layer borders without misleading the reader. In the revised manuscript, we now provide labels on Figure 3- supplement 1 for guidance but without demarcating layer borders*.* We also added extra text in the figure legend to clarify this point and refer the reader to a previous publication from our group that shows back-labeled neurons from neocortical target structures in sagittal sections by using the same injection strategy.

Did you consider doing retrograde injections in CA1 and in RSC/NucAcb to look for double labeled 5a cells?

We did consider doing these experiments. However, because incomplete back-labeling causes under-representation of overlap between the labeled populations we opted for MAPseq and combinatorial viral labeling instead.

Regarding the optogenetic aided circuit mapping experiments: These comprise a large set of recordings which are technically challenging. I was hoping, however, that you might go beyond the characterization of connections to give some functional context to this pathway. In the abstract you state – 'Our results suggest that rather than serving as a relay, deep EC may coordinate hippocampal-neocortical interactions in spatial cognition and memory'. I think this statement requires experiments aimed at understanding the dynamics and magnitude of the connection at the very least.

We are encouraged by the reviewer’s comment as it demonstrates the impact of our study and that our discovery will beget more research on this front to test emerging predictions from our study. To make sure our speculations are not mistaken as conclusions we revised the text in the abstract and result sections to emphasize that our findings are limited to the anatomical properties of deep EC projections. We now discuss our predictions on the functional role of this pathway in the “Ideas and speculation” section.

Reviewer #3:It is not clear whether there are differences in connectivity between Layer 5a and CA1 superficial or deep pyramidal cells.

We thank the reviewer for the suggestion. We now added this analysis to the manuscript and Figure 5 D. We also added a section in the methods that describe our criteria for assigning positional identities to the neurons used in this analysis.

The MAPSeq technique is relatively new and the percentage of shared barcodes between the hippocampus and the telencephalic areas is very high (approaching 99%). Given these high values, more control data would be beneficial.

As we understand it, the reviewer is concerned about false positives. There are three main possible reasons for false positives in these experiments. First is RNA contamination between samples during the dissection. Second is PCR errors. Third is infection of more than one neuron with virus carrying the same RNA barcode sequence. We explain each of these and our control measures in detail in the methods section. Given these control measures, we are confident that the shared barcode values we present here are accurate.

What is the overlap of barcodes across animals or between telencephalic brain areas?

We are unable to understand the value of investigating barcodes that overlap across animals since we do not pool the barcodes across animals. Theoretically there should be minimal overlap since a random selection of barcodes will be expressed in each experiment. We predict that the reviewer might be under the impression that the barcodes were pooled because we present a total number of barcodes across three animals in the legend of Figure 4B. We now present barcode numbers from each of the 3 mice separately as opposed to a total number to prevent misdirecting the readers.

It would be interesting to know whether the layer 5a neurons projecting to the Isocortex also project to the CNU?

This is indeed a very interesting point and the organizational logic of EC layer 5a projections to its telencephalic targets are currently being investigated by our group. However, we believe it is not within the scope of our manuscript as we prefer to keep the focus on establishing the principles of hippocampal projections.

The results on the control brain area shown in the Methods section (line 433) could be moved to the Results section.

We agree with the reviewer that this is an important point that we may not have emphasized enough in the main text. We now make clear in the main text the usage of control tissue and refer the reader to the methods section where control measures are discussed in detail. We also modified the sketch in figure 3 to emphasize the location of where the control tissue was extracted from.

Line 209: Does light stimulation lead to suprathreshold PSP in CA1 pyramidal cells? If so, how confident can we be that the responses observed in interneurons are not due to feedback connectivity between spiking CA1 pyramidal cells and inhibitory neurons? Is the latency sufficiently low to rule out indirect connectivity? It would be beneficial to show the response latency for these responses.

We did not observe suprathreshold responses from pyramidal neurons under our stimulation conditions but some neurons did show large EPSPs (Figure 5 —figure supplement 1E). Nevertheless, we did consider the possibility of indirect activation of the interneurons and addressed this by measuring latencies and pharmacology (presented in the text and figures: Figure 6G,H,I,J and Figure 5—figure supplement 1G,H).

Line 159: The MAPSeq technique is relatively new compared to viral tracing and many readers might not know how it works. I would suggest adding a few sentences explaining this method.

We appreciate the suggestion from the reviewer. We added more explanation on the technique to the relevant section.